# An Outbreak of Japanese Encephalitis Virus in Australia; What Is the Risk to Blood Safety?

**DOI:** 10.3390/v14091935

**Published:** 2022-08-31

**Authors:** Veronica C. Hoad, Philip Kiely, Clive R. Seed, Elvina Viennet, Iain B. Gosbell

**Affiliations:** 1Clinical Services and Research, Australian Red Cross Lifeblood, West Melbourne, VIC 3003, Australia; 2School of Biomedical Sciences, Queensland University of Technology, Kelvin Grove, QLD 4059, Australia; 3School of Medicine, Western Sydney University, Penrith, NSW 2751, Australia

**Keywords:** transfusion, blood safety, infectious diseases, Japanese encephalitis virus

## Abstract

A widespread outbreak of Japanese encephalitis virus (JEV) was detected in mainland Australia in 2022 in a previous non-endemic area. Given JEV is known to be transfusion-transmissible, a rapid blood-safety risk assessment was performed using a simple deterministic model to estimate the risk to blood safety over a 3-month outbreak period during which 234,212 donors attended. The cumulative estimated incidence in donors was 82 infections with an estimated 4.26 viraemic components issued, 1.58 resulting in transfusion-transmission and an estimated risk of encephalitis of 1 in 4.3 million per component transfused over the risk period. Australia has initiated a robust public health response, including vector control, animal control and movement, and surveillance. Unlike West Nile virus, there is an effective vaccine that is being rolled-out to those at higher risk. Risk evaluation considered options such as restricting those potentially at risk to plasma for fractionation, which incorporates additional pathogen reduction, introducing a screening test, physicochemical pathogen reduction, quarantine, post donation illness policy changes and a new donor deferral. However, except for introducing a new deferral to potentially cover rare flavivirus risks, no option resulted in a clear risk reduction benefit but all posed threats to blood sufficiency or cost. Therefore, the blood safety risk was concluded to be tolerable without specific mitigations.

## 1. Introduction

Japanese encephalitis virus (JEV) is a vector-borne flavivirus that causes encephalitis and belongs to the same clade as West Nile virus (WNV), which is a significant blood safety risk [1,2]. There have been a number of case reports of transfusion-transmission of WNV [3] and blood donor testing has been implemented in some endemic areas. In contrast, there has only been a single case report of transfusion-transmitted JEV [4], and blood donor testing is not performed anywhere worldwide. This is despite the fact a study of JE cases estimated that there were 56,847 (95% CI: 18,003–184,525) JE cases and 20,642 (95% CI: 2252–77,204) deaths globally in 2019 [5]. Therefore, total infections would be several million a year.

JEV causes a range of illnesses from asymptomatic to encephalitis that results in death. Most JEV infections in humans do not result in apparent symptomatic illness. The estimated ratio of symptomatic to asymptomatic infections varies from 1 in 25 to 1 in 1000 and depends on both viral and host factors [6]. Overall, symptomatic infections have been estimated to represent less than 1% of infections [2]. JEV leads to a fatal outcome in approximately 20–30% of symptomatic cases. The incubation period for JEV has not been well defined due to limited data. Reported estimates range from 5 to 15 days [7].

JEV is mainly transmitted by mosquitoes of the *Culex* genus. *Culex annulirostris* mosquitoes are the likely regional vector in Australia for JEV [8,9]. The JEV transmission cycle has been previously summarised [2,10,11,12,13,14]. Briefly, wild wading and water birds are major natural JEV reservoirs, including migrating birds like herons and egrets. These birds maintain long-distance spread of JEV. Pigs are also a JEV reservoir and a major amplifying host. Pigs are considered an important host for transmission to humans due to pig–human proximity in farm settings and their higher level viraemia. Humans, cattle and horses are considered accidental and dead-end hosts that are infected when bitten by an infected mosquito, but the brief low level viraemia is insufficient to cause infection in non-infected feeding mosquitoes Australia has a large wild boar population, but little is known about feral pigs and their impact on the transmission cycle and subsequent risk of transmission to humans [9].

Prior to 2022 JEV was not known to have been detected in Australia except in the Torres Strait, Tiwi Islands (which are both islands off northern Australia) and rarely in far northern Queensland [15]. On average, in the last 20 years there has been less than one case per year in Australia [16], all of which were imported except the one from the Tiwi Islands. However, in late February 2022 various state governments announced that JEV had been detected in piggeries in multiple states, and there were human cases of encephalitis of unknown cause [9]. On 4 March 2022, the JEV situation was declared a Communicable Disease Incident of National Significance. The declaration was made under the Emergency Response Plan for Communicable Disease Incidents of National Significance. JEV is a nationally notifiable disease in both humans and animals [17].

Australian Red Cross Lifeblood (Lifeblood) supplies blood and blood products to the Australian community. Given there has only been one case report of transfusion-transmitted JEV disease and humans appear to have brief and low level viraemia, initial rapid risk assessment prior to confirmation of human cases indicated that the risk was likely negligible. In addition, the risk was likely diffuse across a large area of Australia, and therefore it would not be possible to identify people at higher risk of JEV without significantly impacting on sufficiency. Therefore, the initial outcome was to increase surveillance and gain further information on the outbreak. By 31 March there was a total of 32 cases, including suspected and confirmed cases. A formal blood-safety risk assessment was performed to determine if the risk was tolerable, as per the initial assessment. This study describes the rapid risk modelling performed and the associated risk assessment to determine if risk mitigation was feasible or warranted.

## 2. Materials and Methods

### 2.1. Data Source and Study Population

To determine risk management options and estimates of potential donors in the at-risk areas, communication with governments on cases and occupational exposure and publicly available data on cases and infected piggeries were reviewed. Whilst the Australian government has not publicly released detailed case data, there is information on the affected piggeries released by the World Organization of Animal Health and by local media that have been documented in a pre-print study [18]. This was combined with Lifeblood blood donation numbers from 1 January 2021 to 25 April 2022, geospatial information on static and non-static blood donation centres and population estimates to determine an estimate of donors at risk. Standard potential risk management options for blood safety were considered as part of the risk evaluation and risk management, considering risk tolerability principles.

### 2.2. Risk Model Development

A simple deterministic risk model was developed to determine the risk of an encephalitis case in a transfusion recipient over a 3-month period using retrospective data and 32 JEV cases from 1 January 2022 to 31 March 2022.

The risk model included the following data and methods:(1)The asymptomatic to symptomatic ratio was assumed to be 250:1, and therefore, there was an estimated total 8000 cases in Australia in the 3-month period(2)The total population at risk was estimated to be 22.8191 million. Cases at the time of the risk assessment had been detected in the states of Queensland, South Australia, New South Wales (NSW) and Victoria. However, the calculated population at risk also included residents from the Australian Capital Territory (ACT) and Tasmania, given blood collections from those areas are processed in these states and blood to ACT and the state of Tasmania can be from donations collected in Victoria or NSW [19].(3)The number of donors in a 3-month period for clinical components was used for the same blood donor catchment area included in the population at risk. There was a total of 234,212 donors who made an estimated total of 273,965 clinical components that were transfused into recipients. It was assumed that each donor donated a clinical component once and all components were given equal risk of transmission per donor (pooled platelets were given 4 times the risk as pools represent four donors). It is noted that for WNV transfusion-transmission has been reported in all components including leucodepleted (i.e., WBC-reduced) and non-leucodepleted red cells [3].(4)The mean duration of viraemia was estimated to be 4 days. The duration of viraemia is described as brief and low level but, as noted, there is very limited data on the duration of viraemia. Whilst noting that pigs are not dead-end hosts and are expected to have higher and longer duration viraemia than humans, the mean duration of viraemia in pigs was assumed to be 3–5 days [20]. This is supported by a golden hamster model in which no hamsters had JEV viraemia beyond 5 days following inoculation [21]. It was conservatively assumed that 100% of infected donors experienced viraemia even though infection may be controlled in a substantial number by the regional lymph nodes and viraemia may not occur in all infections. Infected donors had a 4/90 (0.044) probability of donating whilst viraemic.(5)A transmission rate of 0.37 was used based on a previous study of lookback follow-up of dengue viraemic donations and the associated transmission rate [22]. It is noted that blood transfusion may not be an efficient transmission route for arboviruses compared to inoculation by mosquitoes [23].(6)The risk in recipients was assumed to be 10-fold higher than the general population. Transfusion recipients are a vulnerable group with high rates of immunosuppression and a mean and median age of 60.6 and 66 years, respectively [24]. There are no reliable estimates of the rate of progression of severe disease of JEV in older or immunosuppressed patients, but small studies indicate outcomes are worse [25]. Similarly, because transfusion-transmission of WNV is better characterised, attempts to find outcomes in transfusion recipients were made as a proxy for JEV. There is one cost-effectiveness study that models the probability of neuroinvasive disease for WNV in transfusion recipients [26]. This used a paper from 1954 where 90 patients with advanced neoplastic disease were intentionally inoculated with WNV via intramuscular injection and 11% developed encephalitis [27]. A large blood usage study demonstrated that approximately 20% was used in haematology and medical oncology combined [28].

Our model conservatively assumes that during the outbreak period, blood donors have an equal risk of being infected as the general population. However, this is unlikely given JEV cases are more likely to be in rural areas without convenient access to donor centres. Therefore, blood donors are likely to represent a subgroup at lower risk of infection than the general population. Overall, the risk model provides conservative risk estimates due to a number of conservative assumptions. The viraemic period is unknown but conservatively assumed to occur in all infections. Immunity was not considered and was set at zero with 100% susceptible for both donors and recipients even though this will change, especially for higher risk donors who are expected to be vaccinated over time.

## 3. Results

### 3.1. Risk Model Results

The number of undetected cases was estimated at 8000 (Table 1). Using the population estimate, the cumulative incidence in donors was 82, which resulted in 4.26 viraemic components being issued, 1.58 resulting in transfusion-transmission and an estimated risk of encephalitis of 1 in 4.3 million per component transfused over the risk period.

### 3.2. Population at Risk Evaluation

One state health department released data on the likely location of acquisition in local government areas (LGAs), which included a total of seven LGAs with confirmed human cases as the suspected place of acquisition and two LGAs with positive mosquitoes [29]. The three positive mosquito samples were from mid and late January tested retrospectively and, with recent samples testing negative, this is consistent with a decreasing risk. Australian Bureau of Statistics (ABS) data indicates there are approximately 100,000 people living in these LGAs. However, visually on a map, whilst some of the involved LGAs are next to each other, they are dispersed over approximately the lower third of the state of NSW, with LGAs without cases in-between (Figure 1). Therefore, given the dispersion and only serious cases of encephalitis are diagnosed, it would be assumed that the risk would be widespread and targeting LGAs with known cases would be ineffective. Approximately 2.8 million live outside of greater Sydney according to 2020 ABS data. If it is assumed that approximately 15% of the area outside of greater Sydney is at risk because the population is more densely concentrated on the coast, that would still be approximately 5.1% of total donors.

Based on information by Yakob et al. (2022) [18] on infected pigs and piggeries, internal blood donation numbers from 1 January 2021 to 25 April 2022 and the coordinates of blood centres (both static and non-static) are presented in Figure 2, demonstrating how these variables potentially interact. Whilst the non-static blood centres only contribute a small proportion of total blood donations compared to the static centres, it is noted that there are a substantial number of donors and donor centres in areas that are considered at risk in the current JEV outbreak including static centres.

Non-official reports note that most reported cases have not had an occupational risk but have been associated with camping, living or travelling to rural areas. Residents of major cities are not considered at risk. However, this does not exclude travel to rural areas for recreation such as camping and at least two reported cases in the media are travel related. Campers and people that travel for recreation outside may be more at risk than people living in the local area given they are likely to camp near water and spend prolonged periods outside. In 2011, another year with significant flooding, another similar flavivirus, Murray Valley encephalitis virus (MVEV), caused an outbreak with 17 cases and three deaths directly from encephalitis in Australia [30]. It is of interest that in the MVEV report, of the 17 cases there were 10 that were residents of the at-risk area, 3 that were non-residents but exposed due to employment and 4 that were tourists. If the undetected milder cases are assumed to be of similar characteristics to the reported cases, any area deferral would only be expected to capture less than 60% of the risk and an employment-based deferral less than 18%, whilst noting that pigs are not amplifying hosts in MVEV so the epidemiology may not be directly applicable.

## 4. Discussion

### 4.1. Risk Evaluation

Lifeblood’s risk tolerability framework indicates that risks should be kept as low as reasonably achievable (ALARA) and gives a numerical level which is ‘acceptable’. For high-impact disease severity agents, the risk may be considered tolerable if it is less than 1 in 1 million (Lifeblood, unpublished). This is consistent with other potentially fatal risks from transfusion. With the current conservative risk estimate during the outbreak from January to March 2022, the risk was calculated as 1 in 4.3 million. To breach the 1 in 1 million threshold, under the same set of conservative assumptions there would need to be 137 cases of encephalitis secondary to JEV in Australia, equating to 34,250 total infections over that same period.

Based on our risk tolerability framework, if the unmitigated risk is in the tolerable risk region and can be further reduced, Lifeblood would introduce donor deferral or product restrictions which do not impact on sufficiency or incur significant cost consistent with the principle of ALARA. However, if the only options to mitigate a risk result in significant costs or sufficiency issues, it may be reasonable to accept the unmitigated risk as a tolerable risk.

JEV is now a communicable disease of national significance, and the Australian government is taking a ‘One Health’ approach. There is an education campaign about mosquito avoidance and mosquito control advice and activities near people, vaccination for those at highest risk and considerable animal industry support and control. The Australian Emergency Management plan for JEV has been activated [31], which includes management and advice of quarantine and movement controls for infected animals, tracing and surveillance plans and advice and guidance to industries on mosquito control and minimising animal exposure.

Public health interventions are expected to reduce the risk for those at highest risk now JEV has been identified. Unlike WNV for which blood donor screening has been implemented in the US, there is an effective vaccine for JEV that is expected to result in a significant risk reduction of asymptomatic viraemia over time if those at highest risk are vaccinated. There are currently two vaccines registered for use in Australia, Imojev and JEspect. While JEspect is an inactivated vaccine, Imojev is a live-attenuated vaccine that is a single dose vaccine and has been demonstrated to result in strong seroprotection [32], although duration in non-endemic areas is less certain. The majority of vaccines secured for Australia in response to this outbreak is the Imojev vaccine. The Australian government has a priority list for vaccination that includes piggery and farm workers and their families living there, personnel who work with mosquitoes and diagnostic and research laboratory workers potentially at risk.

Previous studies of *Culex annulirostris* mosquitoes in the Murray region of Australia demonstrated a peak during February and March when the temperature exceeded 25 degrees, then a rapid decline from April onwards, with no adult activity evident from May to November [33]. At this stage it is unknown whether JEV will become endemic, but it is expected that even if it does, the seasonal risk has peaked and widespread flooding with subsequent water breeding habitats for mosquitoes contributed to the outbreak occurring in 2021–2022 in Australia. In 2011, another year with significant flooding, MVEV caused an outbreak with 17 cases and 3 deaths directly from encephalitis in Australia [30]. MVEV has caused multiple outbreaks in Australia associated with increased rainfall or flooding with the largest recent outbreak in 1974 associated with the 1974 floods with 58 cases [30]. It is noted that despite this, MVEV has not established itself as an endemic virus in south-eastern Australia, despite large previous outbreaks. Notably, MVEV does not have a known pig reservoir. For 2022 at least, it appears the JEV risk peak has passed, with the number of confirmed cases as of 22 June 2022 being 39 [17]. As expected, the risk has decreased due to mosquito activity variation over the cooler months.

### 4.2. Potential Risk Management Options

Potential risk management options are presented for completeness, even if some are not currently warranted or feasible. There are five potential risk management options.

a.Introduce a plasma for fractionation restriction (i.e., do not allow at-risk donors to donate components for direct transfusion)

If any infectious JEV were to be present in a plasma for fractionation pool used to manufacture plasma-derived medicinal products (PDMPs) such as immunoglobulin, the demonstrated efficacy of the specific virus inactivation and removal steps would lead to the conclusion that the final fractionated product is not infectious [34,35]. As a blood safety strategy, there are two options for using a plasma for fractionation restriction to mitigate a risk, an occupational deferral targeting donors in at risk occupations or a geographical deferral targeting donors who reside or visit a risk area.

The risk, therefore, of JEV being transmitted in PDMPs is negligible and hence, similar to other flaviviruses, donors are permitted to donate plasma for fractionation if from an outbreak area and a plasma for fractionation is used as a risk mitigation option allowing donors to continue to donate if well but not to donate fresh components. Dengue and WNV geographical risks are managed through plasma for fractionation restrictions in Australia.

In the past, abattoir workers were restricted to plasma for fractionation predominately due to the concern of Q fever. However, this was removed in September 2017 because of the negligible residual risk and the deferral was not covering the majority of the risk, which was in rural areas outside of abattoirs. Similarly, the majority of the JEV risk would not be covered by a specific occupational deferral for JEV, such as restricting piggery workers to plasma for fractionation. In addition, because piggery workers are being targeted for vaccination, it is expected the residual risk in those occupational groups will decrease over time.

Targeting a geographical area would result in a sufficiency loss. For a geographical area restriction to effectively target the residual risk, it should cover the majority of the risk. Given diagnosed cases only represent a small proportion of total cases, it would likely be almost completely ineffectual to target areas such as LGAs where there are known diagnosed cases as there are likely to be other areas of equivalent risk due to undiagnosed cases, as demonstrated by the case detection dispersion in NSW (Figure 1). Using known case data (unlike dengue which has a higher symptomatic rate) to target geographical restrictions is not feasible with a risk that is likely diffuse, and the geographical range has not been well documented. There are also the ongoing operational issues of applying deferrals to previously untargeted areas when cases are reported for the first time, and the development of criteria for removal of restrictions.

In addition, it would be necessary to develop guidelines to respond to the detection of JEV in mosquitoes or animals in an area when there are no known human cases, as was the case in NSW where there were two positive mosquito areas without known positive human cases.

Consideration would need to be given to any restriction if it would be applied to travel or residence or both. For dengue outbreaks, a map of the area is used, and each donor is asked if they have travelled there in the last 4 weeks. This would be difficult and for donors, if constantly changing, would be difficult to implement operationally.

In the context of a residual risk that was intolerable and likely to result in multiple fatalities of transfusion recipients, a large area plasma for fractionation restriction may be warranted as a temporary measure. However, with the current residual risk this does not apply to the JEV risk considering risk tolerability principles.

b.Introduce a screening test

There are no licensed tests for screening blood donors for JEV. It is noted that by the time a person tests positive by serology they would be expected to have cleared the viraemia or be non-infectious. Therefore, a molecular test for JEV RNA would be the only option. A large pharmaceutical company would need to be committed to evaluate and push for a class IV IVD license for registration in Australia for a specific JEV NAT test. Given the future of JEV in Australia is uncertain and, in a global context, the small market share of blood donors in Australia, it is very unlikely that a donor NAT screening test for JEV would be commercially attractive. In addition, given the current estimated residual risk, it is not warranted and almost certainly not cost-effective. There is a reported probable case of TT-WNV that occurred with minipool testing [36]. Given there is limited viral load data, any optimal testing strategy would be difficult to determine in terms of pooled testing versus individual NAT testing and evaluation would also be difficult.

c.Physicochemical pathogen reduction for fresh blood components

There is no licensed pathogen reduction technology (PRT) for fresh blood components available in Australia. The Theraflex MB Plasma and Platelets Systems (MacoPharma, Tourcoing, France) effectively inactivates JEV [37]. Specific data relating to JEV and the efficacy of other commercially available PRTs is not available. However, the INTERCEPT Blood System (Cerus Corporation, Concord, CA, USA), Mirasol Pathogen Reduction System (Caridian BCT Biotechnologies, Lakewood, CO, USA) and S-303 Pathogen Reduction System for red blood cell products (Cerus Corporation, Concord, CA, USA) effectively inactivate the related flaviviruses, West Nile and dengue viruses; the INTERCEPT Blood System also inactivates Zika virus. Therefore, these PRTs may also effectively inactivate JEV [38,39,40,41,42,43,44].

There is no licensed pathogen reduction system for red cells globally or in Australia. In the risk assessment period, red cells consisted of 63.8% of total components or 49.5% if pooled platelets are considered equivalent to 4 donor exposures as assumed in the risk assessment. Therefore, even if pathogen reduction was a feasible option, it would not currently be suitable to reduce the majority of the residual risk. Previous assessments have concluded that pathogen reduction is not cost effective in Australia.

d.Quarantine or enhanced post donation notification or recall

A few years ago, Lifeblood strengthened its post donation notification process, and a donor is asked to notify us if they develop a cough, cold, diarrhoea or other infection within a week of donating or are diagnosed or hospitalised with a serious infection within 2 months of donating. However, given the very low symptomatic rate of JEV this is not expected to impact the JEV risk. Currently if a donor reports a fever post donation, components are recalled if it occurs on day zero to two with no recall after day three. Increasing the length of time for which blood components are recalled for a non-specific illness would not decrease the blood safety risk but result in an increase in unnecessary recalls. Therefore, post donation illness reporting is not expected to modify JEV risk to blood safety. Similarly, given donors are viraemic mostly when asymptomatic, quarantine (i.e., holding donations for a period to identify subsequently symptomatic donors) is not an effective strategy.

e.Change donor selection criteria

Lifeblood does not have a specific deferral for JEV, but it does have a WNV deferral. Given the risk is from asymptomatic donors, this lack of a specific deferral will not change the risk profile. However, JEV is now expected to be more common than WNV in Australia. Therefore, we have recommended a flavivirus deferral is implemented to ensure that in the event of post donation notification there is an appropriate deferral period and a specific instruction to recall. Currently, as previously noted the fever recall is day 0 to day 2 and there is an encephalitis recall which is within 4 weeks of illness onset. There is a potential gap a case with less severe illness is diagnosed i.e., fever only. A generic flavivirus entry would also ensure there are recall instructions for other flaviviruses not covered by a specific entry. However, whilst good practice, this deferral introduction will not significantly alter the residual risk.

f.Accept the risk

Given,

the residual risk of a severe outcome is negligible;the risk is decreasing over winter;public health initiatives will modify the risk to humans;the other options are not expected to decrease the risk significantly or are not available and expected to result in either increased cost or sufficiency implications which are not commensurate with the risk.

Then, this risk assessment demonstrates the risk is negligible, and therefore this new risk should be currently accepted as tolerable.

Because the assumptions in the model are based on limited data, it is difficult to validate the numerical risk assessment. However, we believe our careful evaluation of the risk management options is sound. A viraemia prevalence study with recipient follow up would be the most effective way to validate the model. However, given the point estimate of the risk of a viraemic blood donor at the peak of the outbreak was estimated to be approximately 1 in 64,000, any study would need to be very large to determine an accurate prevalence to inform the risk assessment. Given the absence of recorded cases in blood donors or recipients during this outbreak (in keeping with our model assumptions) and the absence of a significant risk signal internationally, this is considered not commensurate with the risk.

There is significant uncertainty with the outbreak in Australia and how it will progress in the future. Therefore, whilst the risk is currently estimated as tolerable, Lifeblood continues to monitor the risk with enhanced surveillance data and will adjust our risk assessment if emerging information would materially alter our estimation. To further evaluate the outbreak risk in Australia, Lifeblood is currently performing a serosurvey in blood donors with other stakeholders that will provide an estimate of exposure in blood donation centres considered at higher risk, using centres outside of the outbreak area as a control. However, this may be difficult to extrapolate to a blood safety risk given travel, vaccination and the fact that it is a cumulative prevalence.

## 5. Conclusions

Despite considerable uncertainty of the risk to public health and what this will look like in the future, the current outbreak, with conservative modelling, is considered a negligible risk. Potential risk mitigation options are not currently feasible or warranted.

Based on our model there would need to be approximately 45 cases of JEV encephalitis a month to breach the 1 in 1 million threshold (representing 32,450 undetected cases a month), and these numbers would be unsustainable for long periods given infection gives lifetime immunity and are therefore, over time, less likely to occur. In addition, with lower density of mosquito vectors over winter the risk has decreased.

A robust public health response, including vector control, animal control and movement, surveillance and vaccination, is anticipated to impact on the risk as further information is elicited and the ability of a one health response to prevent large, repeated outbreaks from occurring is interlinked with the blood safety risk assessment. The risk to blood safety is associated with risk of viraemic donors which is dependent on the ability of a coordinated response to limit transmission. Whilst it has been noted that the identification of JEV in Australia has many analogies to the introduction and dissemination of WNV in the United States [45], where WNV blood donor screening is required, population density differences, the existence of an existing effective vaccine and the association with severe flooding are key differences. How JEV will persist in mainland Australia is uncertain. Lifeblood continues enhanced surveillance and information gathering and will reassess the risk as required. However, the risk is currently accepted as tolerable, given it is negligible and the conclusion is clear that mitigations do not have a clear risk reduction benefit but pose threats to either sufficiency or cost.

## Figures and Tables

**Figure 1 viruses-14-01935-f001:**
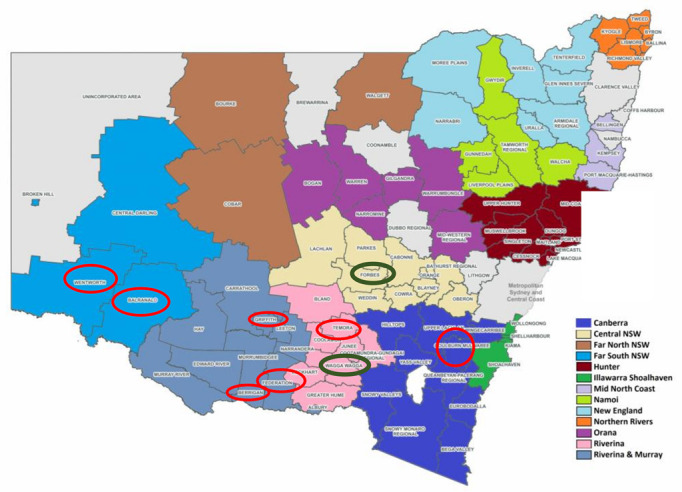
LGAs* with confirmed human cases circled in red and positive mosquito populations circled in green. (*Map modified from Joint Organisations—Office of Local Government NSW (https://www.olg.nsw.gov.au/programs-and-initiatives/joint-organisations/#:~:text=NSW%20boasts%20a%20network%20of,service%20delivery%20to%20regional%20communities, accessed on 27 June 2022)).

**Figure 2 viruses-14-01935-f002:**
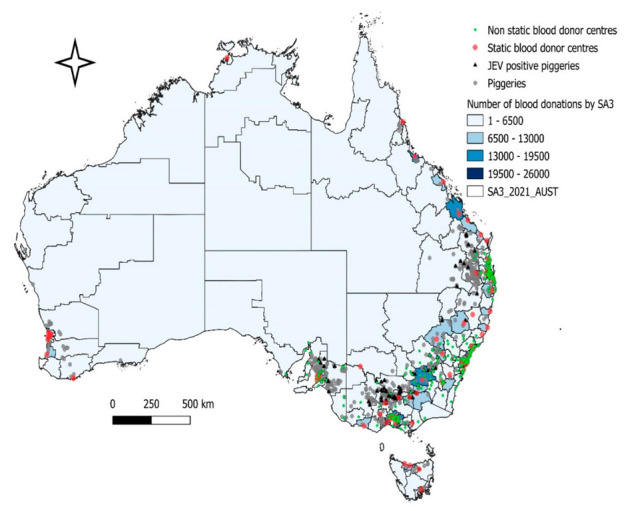
Blood donation centres, number of blood donations (from 1 January 2021 to 25 April 2022) and piggery locations in Australia.

**Table 1 viruses-14-01935-t001:** Estimate of number of symptomatic cases of Japanese encephalitis secondary to blood transfusion.

Estimated Number of	Results
Cases in risk area (diagnosed cases × 250)	8000
Risk per person (cases/population)	0.0003506
Number of donors and components	234,212; 273,965
Cumulative prevalence in donors (Risk × donors)	82
Number viraemic donors and components (0.044 probability)	3.65, 4.26
Risk of transmission (viraemia x transmission	1.58
Japanese encephalitis TT cases (case/undiagnosed × 10)	0.063
Risk of Japanese encephalitis cases (TT cases/components)	1 in 4.3 million

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
