# Peer review of "An Outbreak of Japanese Encephalitis Virus in Australia; What Is the Risk to Blood Safety?"

_viruses, 2022, doi:10.3390/v14091935_

Round 1
Reviewer 1 Report
Dr. Hoad group showed A model of Blood Safety Risk Assessment for Emergence of Japanese Encephalitis Virus in Australia. They predicted that the cumulative estimated incidence in donors was 82 infections with an estimated an estimated risk of encephalitis of 1 in 4.3 million per component transfused over the risk period in Australia in 2022.
Major comments:
1. As authors mentioned 234,212 donors attended during over a 3-month JEV outbreak period, but did not discover the case number confirmed by clinical lab diagnosis. How to evaluate the proposed model?
2. Authors should discuss the linkage of Blood Safety Risk Assessment model with public health responses, such as vector control, animal control and movement, and surveillance.
3. The title of the manuscript was not specific to the context, and similar to some of prior publications about a JEV outbreak in Australia in 2022.
Author Response
- As authors mentioned 234,212 donors attended during over a 3-month JEV outbreak period, but did not discover the case number confirmed by clinical lab diagnosis. How to evaluate the proposed model?
Thank you for your review. We have inserted and changed the below paragraph:
Because the assumptions in the model are based on limited data, it is difficult to validate the numerical risk assessment. However, we believe our careful evaluation of the risk management options is sound. A viraemia prevalence study with recipient follow up would be the most effective way to validate the model. However, given the point estimate of the risk of a viraemic blood donor at the peak of the outbreak was estimated to be approximately 1 in 64,000, any study would need to be very large to determine an accurate prevalence to inform the risk assessment. Given the absence of recorded cases in blood donors or recipients during this outbreak (in keeping with our model assumptions) and the absence of a significant risk signal internationally, this is considered not commensurate with the risk.
- Authors should discuss the linkage of Blood Safety Risk Assessment model with public health responses, such as vector control, animal control and movement, and surveillance.
We have made this clearer in text:
A robust public health response including vector control, animal control and movement, surveillance, vaccination is anticipated to impact on the risk as further information is elicited and the ability of a one health response to prevent large, repeated outbreaks from occurring is interlinked with the blood safety risk assessment. The risk to blood safety is associated with risk of viraemic donors which is dependent on the ability of a coordinated response to limit transmission. Whilst it has been noted that the identification of JEV in Australia has many analogies to the introduction and dissemination of WNV in the United States [46], where WNV blood donor screening is required, population density differences, the existence of an existing effective vaccine and the association with severe flooding are key differences. How JEV will persist in mainland Australia is uncertain. Lifeblood continues enhanced surveillance and information gathering and will reassess the risk as required. However, the risk is currently accepted as tolerable, given it is negligible and the conclusion is clear that mitigations do not have a clear risk reduction benefit but pose threats to either sufficiency or cost.
- The title of the manuscript was not specific to the context, and similar to some of prior publications about a JEV outbreak in Australia in 2022.
We have changed the title as suggested
Reviewer 2 Report
The references are appropriate. The assuptions in this paper are reasonable based on the available data. The conclusions are quite reasonable.
Line 378 "periods of time" is tautology. "long periods" is sufficient.
Author Response
We have changed as suggested and thank you for your positive review
Round 2
Reviewer 1 Report
Authors had well responded to my comments in the revision.